# Mouse lung automated segmentation tool for quantifying lung tumors after micro-computed tomography

**Mary Katherine Montgomery[1]☯, John David[1]☯, Haikuo Zhang[2], Sripad Ram[3], Shibing Deng[4], Vidya Premkumar[1], Lisa Manzuk[1], Ziyue Karen Jiang[1], Anand Giddabasappa[1] ***

**1** Comparative Medicine, Pfizer Inc., La Jolla, CA, United States of America, **2** Oncology Research Unit, Pfizer Inc., La Jolla, CA, United States of America, **3** Drug Safety Research Unit, Pfizer Inc., La Jolla, CA, United States of America, **4** Early Clinical Development, Pfizer Inc., La Jolla, CA, United States of America

☯ These authors contributed equally to this work.
* anand.giddabasappa@pfizer.com

**Data Availability Statement:** All relevant data are within the paper and Supporting Information files.

**Funding:** This project was funded by Pfizer, Inc. Pfizer Inc. provided financial support in the form of

## Abstract

Unlike the majority of cancers, survival for lung cancer has not shown much improvement since the early 1970s and survival rates remain low. Genetically engineered mice tumor models are of high translational relevance as we can generate tissue specific mutations which are observed in lung cancer patients. Since these tumors cannot be detected and quantified by traditional methods, we use micro-computed tomography imaging for longitudinal evaluation and to measure response to therapy. Conventionally, we analyze microCT images of lung cancer via a manual segmentation. Manual segmentation is time-consuming and sensitive to intra- and inter-analyst variation. To overcome the limitations of manual segmentation, we set out to develop a fully-automated alternative, the Mouse Lung Automated Segmentation Tool (MLAST). MLAST locates the thoracic region of interest, thresholds and categorizes the lung field into three tissue categories: soft tissue, intermediate, and lung. An increase in the tumor burden was measured by a decrease in lung volume with a simultaneous increase in soft and intermediate tissue quantities. MLAST segmentation was validated against three methods: manual scoring, manual segmentation, and histology. MLAST was applied in an efficacy trial using a Kras/Lkb1 non-small cell lung cancer model and demonstrated adequate precision and sensitivity in quantifying tumor growth inhibition after drug treatment. Implementation of MLAST has considerably accelerated the microCT data analysis, allowing for larger study sizes and mid-study readouts. This study illustrates how automated image analysis tools for large datasets can be used in preclinical imaging to deliver high throughput and quantitative results.

## Introduction

Lung cancer is the leading cause of cancer deaths worldwide. Each year, 2.1 million people are diagnosed with the disease and 1.8 million die from it [1]. Lung cancer is broadly classified as

research materials and salaries for authors [MKM, JD, HZ, SR, SD, VP, LM, KJ, AG], but did not have any additional role in the study design, data collection and analysis, decision to publish, or preparation of the manuscript. The specific roles of these authors are articulated in the 'author contributions' section.

**Competing interests:** All authors are or were full- or part-time employees of Pfizer, Inc. This does not alter our adherence to PLOS ONE policies on sharing data and materials.

non-small cell lung cancer (NSCLC), the most prevalent form which makes up 85% of the patients, or small cell lung cancer (SCLC), which accounts for the remaining 15% of lung cancer patients [1]. There has been tremendous progress in the treatment of lung cancer with targeted therapies in the last two decades [2] and very recently with immuno-oncology agents [3].

Lung tumors are driven by mutations in oncogenic or tumor suppressor genes resulting in gain or loss of function, respectively [4–6]. The most common oncogenes that carry driver mutations include EGFR, Kras, ALK and tumor suppressors p53 and RB1. Discovery of these driver mutations has led to novel targeted therapies which have shown significant promise in selected patients [7]. Genetically engineered mouse (GEM) models of lung cancer have played important roles in understanding the driver mutations and disease pathology as well as in assessment of novel drug candidates prior to clinical evaluation [8, 9].

Unlike traditional subcutaneous xenograft models, GEM lung tumor models present a unique challenge because tumors spontaneously develop within the lungs just as they do in patients. Quantifying treatment effects in these internal tumors requires imaging as they cannot be measured with calipers antemortem. Computed tomography (CT) imaging is used clinically for standard assessment and grading based on Response Evaluation Criteria in Solid Tumors (RECIST) criteria [10]. Advances in micro-computed tomography (microCT) imaging technology have led to its use in evaluating lung GEM models non-invasively [11, 12].

Previously we demonstrated proof of concept in using microCT technology to evaluate the efficacy of targeted therapies in NSCLC GEM models [13]. There have been multiple efforts by other teams to evaluate lung tumors in mice [14, 15]. Traditional methods of manual analysis generally include qualitative assessment where a radiologist or researcher would assign an ordinal score on a system similar to those utilized in pathology [16], or quantitative analysis via manual segmentation of lung tumors with software tools such as "paint" or "lasso" (Vivo-Quant™, Invicro, Boston, MA) ideally with costly contrast-enhancement [12, 13]. In a 3D lung scan, this process is time consuming and prohibits interim reads of large studies. In addition, the results can introduce intra/inter-analyst variability. To address these challenges, we developed the mouse lung automated segmentation tool (MLAST). MLAST automatically segments lung-field microCT scans into tissue types by density and allows for quantitative tracking of lung air volume throughout disease progression in a variety of murine lung tumor models. MLAST has been validated against manual scoring, manual segmentation, and histology, and has demonstrated an ability to accurately characterize loss of lung space in diseased animals.

## Methods

### Animal model

All animal procedures were performed as described in the animal use protocols approved by Pfizer's Institutional Animal Care and Use Committee. The non-small cell lung cancer GEM mice KP ($Kras^{G12D-LSL/+}/p53^{fl/fl}$) and KL ($Kras^{G12D-LSL/+}/Lkb1^{fl/fl}$: B6.Lkb1-B6.129S4-Krastm4-Tyj/J) were purchased from Jackson Laboratories (Bar harbor, ME). Lung tumors were induced as previously described by DuPage M et al., 2009 [17]. Briefly, 8–12 weeks old KP or KL mice were induced with $1 \times 10^6$ PFU adeno-virus (Ad5CMVCre, University of Iowa) intranasally under 3% isoflurane anesthesia. Health checks of the animals were performed weekly. After six weeks post-virus delivery, microCT imaging was initiated to evaluate the lung tumor burden. During every imaging session the animals were anesthetized with 1.5 to 3% of isoflurane and maintained under isoflurane anesthesia throughout the imaging session. Since the animals had tumors internally (in lungs) they were euthanized at the end of the study or based on the body condition scoring method [18], per the Pfizer IACUC's guidelines. Euthanasia was

performed by $CO_2$ inhalation and cervical dislocation as a secondary method. When lungs were collected for immunohistochemistry, the animals were euthanized by $CO_2$ and inflated with formalin. The radiation dose was <1 μSv per scan, which is several orders of magnitude less than a therapeutic dose [19]. Radiation exposure from the scans was therefore deemed to have no impact on the observations of this study.

To validate non-contrast segmentation of lung tumors with MLAST, KL tumor bearing mice (n = 10) were longitudinally imaged twice at 4 time-points, once pre-induction, and at weeks 4, 8, and 12 post-induction. The first scan was non-contrast followed by a contrast-enhanced scan.

For contrast-enhanced imaging, animals were injected IV with 100 μl of Viscover ExiTron Nano 12,000, an inorganic iodine nanoparticulate (Miltenyi Biotec, Bergisch-Gladbach, Germany), and were imaged 15 minutes post-injection using parameters described previously [13]. MLAST was used to segment all non-contrast scans. Manual segmentation was performed first on the non-contrast scans and then on the contrast scans by a single analyst who was blinded to time-point and scan pairing. The volumes segmented from the non-contrast scans by both manual segmentation and MLAST were compared to the volumes manually segmented from the contrast-enhanced scans by calculating the Pearson's correlation coefficient. The non-contrast lung segmentations were also compared with a Sorenson-Dice coefficient, as is commonly used in image segmentation validation efforts [20].

MLAST was utilized to evaluate efficacy of a drug in a KL-GEM model of lung cancer. Non-contrast microCT imaging was performed biweekly after 6 weeks post-virus induction until tumors reached an advanced score between 2 to 2.5. Animals were dosed with the drug via oral route for 21 days. The pretreatment scans were compared with post-treatment images by manual scoring and MLAST, as described below.

## microCT acquisition

Images were acquired with a SkyScan 1278 (Bruker Corporation, Billerica, MA) using a 0.5 cm Al filter at 918 μA, 50 kV, 100 μm resolution, and a step size of 0.5˚ per projection for 360˚ with scan time of ~2 minutes. Images were Feldkamp reconstructed with an isotropic 100 μm voxel, ring artifact correction, a 12% beam hardening correction, Gaussian smoothing with a kernel size of 2, and a pixel matrix of approximately $230 \times 310 \times 185$ (NRecon 1.7.1.0, Bruker, Billerica, MA).

## microCT data analysis

microCT scans were evaluated by three different methods: manual scoring, manual segmentation and automated analysis using MLAST. Manual scoring was performed by a blinded scientist using Amira 6.3 image analysis software (ThermoFisher Scientific, Waltham, MA). Scientists viewed scans in 3D using different orientations to look for tumor nodules, and assigned each scan a score on a 0–3 ordinal scale. A score of 0 indicated a scan with no visible tumors. A score of 1 was assigned to a scan with one or two small tumor nodules. A score of 2 indicated a scan with mid-size to large tumors that were beginning to fuse together. Finally, a score of 3 was assigned to any scan in which a large coalescing mass of tumor occupied more than half the lung space (Fig 1).

Manual segmentation was also performed by blinded scientists using Amira 6.3 software with the "paint" tool and interpolation every two slices to classify the lungs, heart, and tumors in the scan. Vasculature was not separately classified due to the inability to segment tumor from vessels in non-contrast microCT scans. The caudal bound was set at the diaphragm and

**Fig 1. Representative images showing manual qualitative scores.** A score of 0 contains no signs of tumor mass. The contrast in the lung space is derived from blood vessels and heart tissue. A score of 1 shows one or two small nodules within the lung field. A score of 2 shows more nodules that are beginning to coalesce, and a score of 3 shows a large coalescing mass that has infiltrated more than half the lung space.

was excluded from segmentation. The cranial bound was set at the tracheal bifurcation. The average number of slices was 93.0 +/- 1.50.

## Automated analysis using MLAST

The MLAST algorithm was written in Matlab R2017b, and utilized the Image Processing, Statistics and Machine Learning, and Parallel Computing toolboxes. The algorithm estimated the outer limits of the thoracic cavity and automatically separated the voxels into different tissue types based on density: soft tissue, lung, and intermediate. The soft tissue is inclusive of tumors, esophagus, heart, and vasculature; these tissues are binned together due the similar density on microCT. The intermediate classification is one of lower density on the microCT and represents a mix of motion artifacts of respiration in small nodules and non-solids lesions with either an acinar or ground-glass appearances [21]. The non-diseased lung segmentation was then expressed as a percentage of the total thoracic cavity and tracked over longitudinal scans for a given animal. The non-diseased lung area had significant decreases in the percentage due to space occupying lesions. The specific steps are demonstrated in Fig 2 and described in detail below.

First, the thoracic cavity was identified using the ribcage in a similar approach to that described in Barck et al., 2015 [14]. Scans were thresholded to identify high-density bone and the largest connected 3D object was identified. Three slices were stacked to smooth the transition between slices (an improvement on Barck *et al.*'s method), and the bones were used to create a mask outline of the thoracic cavity. In each 2D axial slice, groups of connected bone pixels were identified as individual ribs or spine, and the innermost points of these bones were reordered clockwise by the angle between their centroids and the centroid of the entire ribcage (Fig 2A). A spline function was then used to interpolate between these points, producing a rough outline of the interior of the ribcage, and therefore the exterior of the thoracic cavity. To improve MLAST performance in scans with minor motion artifact, two iterations of this approach were included. In the first, all points in the outlines of the rib regions were used to draw the initial mask of the thoracic cavity. The interior of this mask was then used as the centroid for the next iteration of the approach, in which only the innermost 30% of the rib points were used to draw the outline of the new mask. This approach ensured a more stable mask and helped to prevent the accidental inclusion of the intercostal muscles in the mask of the thoracic cavity.

Next, the data was automatically segmented using a one-dimensional implementation of the unsupervised machine learning algorithm k-means clustering (MathWorks, Natick, MA), which broke voxels into clusters according to density. The k-means algorithm used a k+

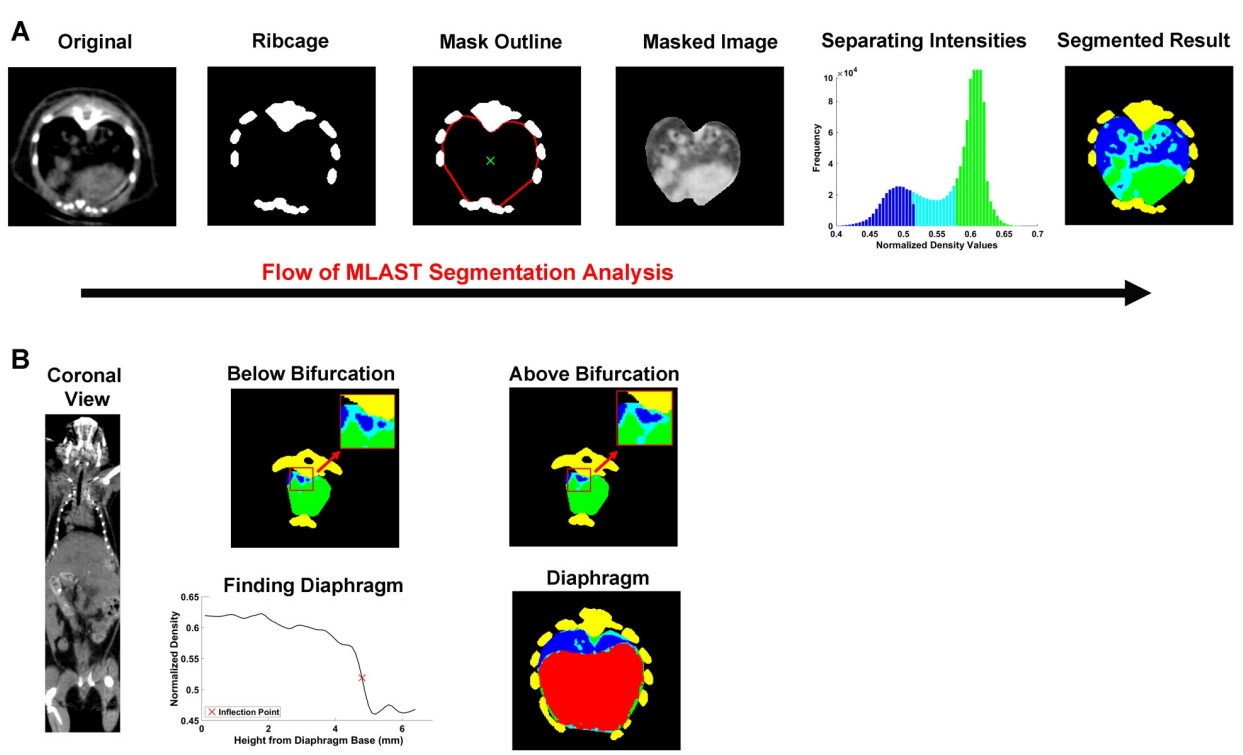

**Fig 2. Schematic of MLAST analysis. (A)** Depiction of MLAST's segmentation algorithm. The original microCT 2D image contains the densities of various tissues observed in a thoracic scan. The ribcage provides a high contrast for thresholding. The interior points of the rib regions are used to create a mask of the thoracic cavity. The densities of the voxels within the masked image are segmented with a k-means clustering algorithm and the intensities are classified into 3 tissue types: soft tissue (green), lung (blue), and intermediate (cyan). The final segmented result is shown in 2D and applied to all slices in the z-stack. **(B)** Depiction of how MLAST detects the cranial and caudal boundaries of the thoracic cavity. The cranial boundary is determined by the bifurcation of the trachea. Below the bifurcation, the two tracheal regions (colored blue and shown in inset) are separate. Tissues above the bifurcation, where the two regions are no longer separable, are excluded from the final segmentation. On the caudal end, the diaphragm is segmented based on density changes in the z-trace of each voxel. The resulting diaphragm (red) is removed from the thoracic counts of soft tissue, lung, and intermediate.

\+ seeding to set initial cluster centroids at 3 different locations on the spectrum of densities seen in the image. Voxels were assigned to the cluster with the nearest centroid, and then the centroids were updated to the center of all densities assigned to the cluster. This process was repeated until either the cluster assignments stopped changing or the maximum number of iterations (100) was reached. The cluster with the lowest-density centroid was assigned to lung, the cluster with the highest-density centroid was classified as soft tissue, and the middle cluster was declared intermediate (Fig 2A).

Finally, the cranial and caudal cutoffs were determined as demonstrated in Fig 2B. Similar to the method described in [14], the z-slice where the tracheal region split in two was used as the cranial cutoff. The top of the diaphragm at the heart-diaphragm interface was identified as the point where the thoracic cavity contained the highest percentage of lung. MLAST then segmented the 3D shape of the diaphragm between the caudal end of the scan and the heart-diaphragm interface using the inflection point of each pixel's intensity. All tissue identified as diaphragm was removed from the thoracic cavity and therefore from the volumes of the segmented tissues. This allowed MLAST to segment tumors located adjacent to the diaphragm and to decrease potentially confounding effects in the case of a large tumor at the heart-diaphragm interface. Fig 3 illustrates the various MLAST segmented slices of the lung from diaphragm to the tracheal bifurcation. The differential tissue densities are color-coded to represent different

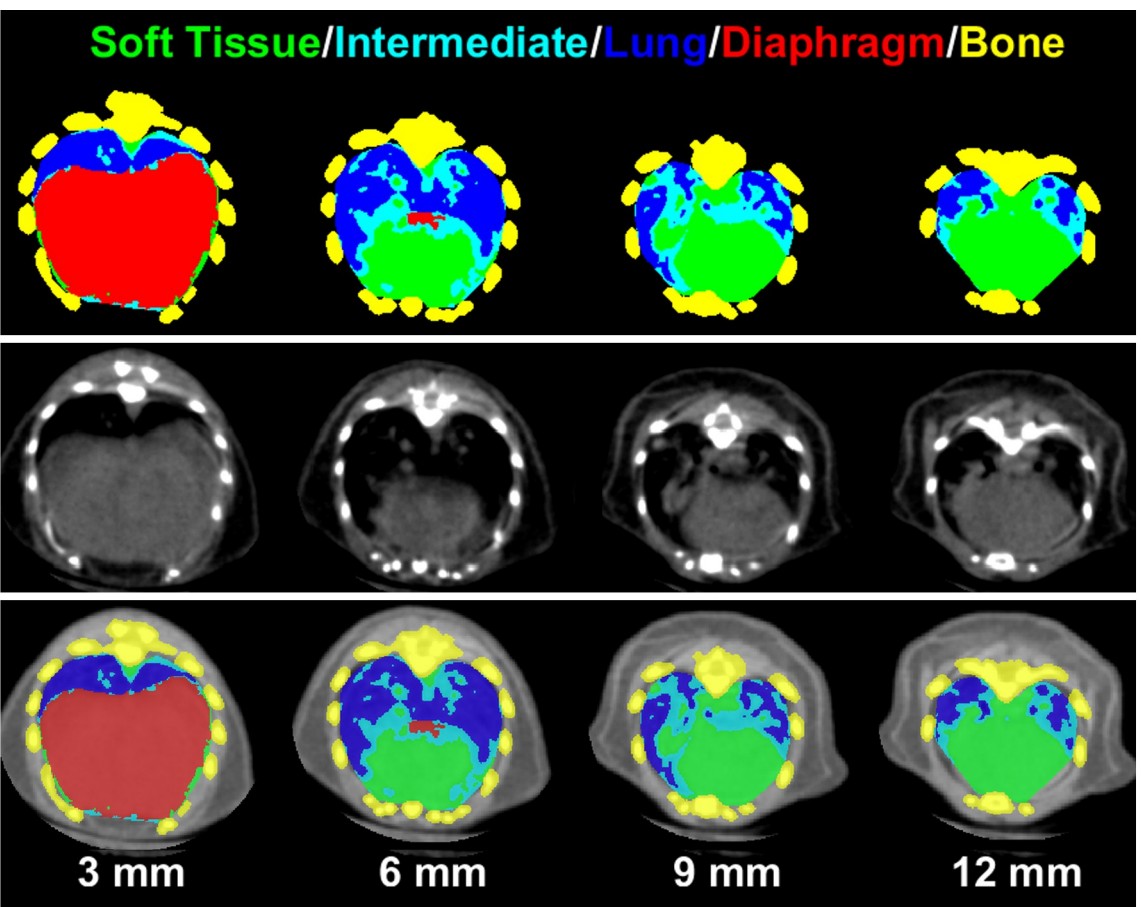

**Fig 3. Representative slices of lung from a tumor-bearing mouse showing the resulting segmentation from the MLAST algorithm at various levels (3, 6, 9 and 12 mm) above the base of the diaphragm.** Top panel is MLAST only, middle panel is microCT only and the bottom panel is the overlay of MLAST and microCT images. The tissue labels for MLAST segmentation are color-coded for visual purposes: soft tissue (green), intermediate (cyan), lung (blue), diaphragm (red), and the mutlislice bone mask (yellow).

tissues (lung, soft tissue, intermediate, diaphragm and bone) as determined by MLAST. The MLAST results are expressed in terms of the percentage of the thoracic cavity made up of lung. Decreases in this lung percentage can be attributed to increases in tumor burden.

## Histology

Tumor bearing animals were euthanized via $CO_2$ inhalation. Lungs were perfused and incubated for 48h in formalin before embedding in paraffin. The FFPE blocks were sectioned 4 μm thick and stained with hematoxylin and eosin (H&E). Slides were scanned using a Leica/Aperio AT2 whole slide digital scanner at 20X magnification. Images were analyzed using Visiopharm 2017.7. First the lung tissue was detected, and the total lung area was calculated. Tumor lesions were then detected using a machine learning algorithm based on the random forest classifier. Results from Visiopharm were exported to Excel.

## Statistical analysis

The MLAST scores, expressed as a percentage, were compared against the manual scores using Spearman's rank correlation, and the outputs for each tumor category (1–3) were compared

against the tumor free scans (category 0) with a series of Student's t-tests using a Bonferroni-corrected alpha level of 0.0056.

Volumes segmented by MLAST were compared to those segmented manually (in both contrast and non-contrast scans) via linear regression with Pearson's correlation. MLAST segmentations of the lung field were compared more directly to manually segmented non-contrast scans with a Dice similarity coefficient. Spearman's rank correlation coefficient was used to compare MLAST measurement of % tumor burden with the % of tumor metastasis quantified from H&E images of lung sections.

Logistic regression was used to assess the drug's treatment effect on the change in manual score from baseline to post treatment. In general, a proportional odd model could be used when more than two unique values of score change were observed. However, in this case only two unique values of score change was observed (the score was either unchanged or increased by 1), so a logistic regression was used. To evaluate the treatment effect on % lung calculated from MLAST, analysis of covariance (ANCOVA) was applied with baseline % lung as a covariate.

## Results

### Comparison of MLAST to qualitative manual scoring

Previously we used manual segmentation and contrast-enhanced microCT imaging to evaluate mouse models of lung tumor and response to therapy [13, 22]. Although these methods are quantitative, they are low-throughput and contrast is costly. Thus, we investigated alternative approaches to increase throughput and minimize variability within and between analysts. Manual scoring by an image analysis scientist was adopted to evaluate each animal's lung tumor burden. We chose 446 scans from multiple studies on which to perform manual scoring. Samples with no lung tumors were assigned a score of 0, whereas lungs with very high tumor burden were scored as 3 (Fig 1, described in methods).

The MLAST algorithm was applied to 20 scans with a score of 0 to evaluate non-tumor bearing lung fields. In these scans, the thoracic cavity constituted ~43% lung, ~37% soft tissue, and ~19% intermediate tissue (Table 1). A representative pseudo-colored image panel and 3D flythrough of the thorax from the caudal to the cranial end is shown in Fig 3 and S1 Video. Later, MLAST was applied to all 446 scans which were manually scored. The MLAST outputs were then compared against the manual scores using a Spearman's rank correlation, and the outputs for each tumor category (score 1–3) were compared against the tumor free scans (score 0) with a series of Student's t-tests with a Bonferroni-corrected alpha level of 0.0056. Scans with higher manual scores showed a decrease in lung space and a simultaneous increase in the proportion of soft tissue and intermediate tissue (see Fig 4A and Table 1). The differences between the manual scores 1 and 2 were statistically significant for all 3 tissue types: lung, soft tissue, and intermediate tissue, whereas the differences between the manual scores 2

**Table 1. Validation of MLAST with manual scoring (related to Fig 4A).**

| Score | Soft Tissue | Lung | Intermediate |
|---|---|---|---|
| 0 (n = 20) | 37.34 +/- 0.47 | 43.36 +/-1.15 | 19.28 +/- 1.21 |
| 1 (n = 127) | 37.62 +/- 0.36 | 41.85 +/- 0.48 | 20.52 +/- 0.45 |
| 2 (n = 202) | 40.09 +/- 0.34 | 34.44 +/- 0.49 | 25.45 +/- 0.39 |
| 3 (n = 97) | 48.56 +/- 0.81 | 25.80 +/- 0.69 | 25.62 +/- 0.56 |

Table showing mean +/- standard error of MLAST results expressed as % of thoracic cavity for scores 0, 1, 2, and 3.

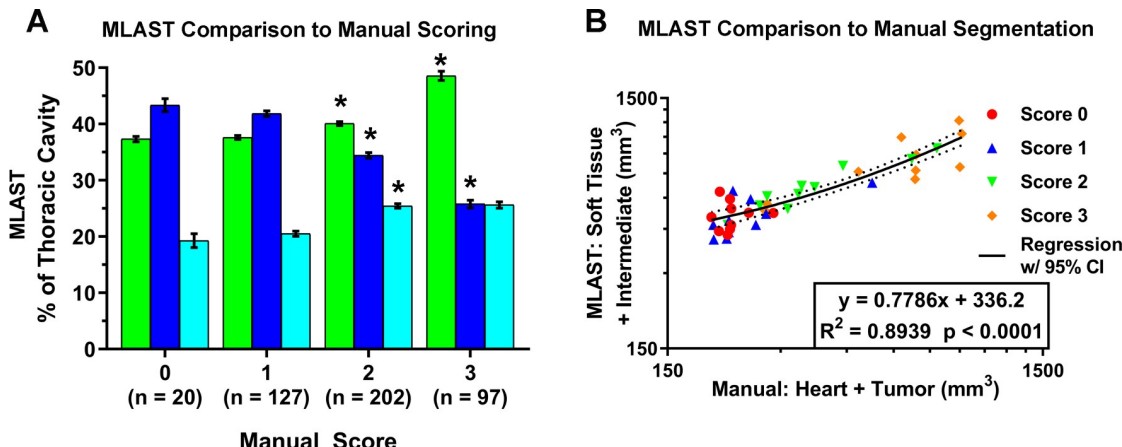

**Fig 4. Comparison of MLAST with manual scoring system and manual segmentation. (A)** Results comparing MLAST values (% thoracic cavity) to manual scores on a 0–3 scale. MLAST values represent soft tissue (green bar), lung (blue bar) and intermediate density (cyan bar). The number of samples that corresponded to manual scores 0–3 is represented in Y-axis. Error bars are standard error of mean and * indicates statistical significance by Student's t-test at a Bonferroni-corrected significance level of α = 0.0056. **(B)** Comparison of the three methods: manual scored samples, manually segmented volume and MLAST values. Linear regression analysis of automatically segmented thoracic tissue (intermediate + soft tissue) by MLAST vs manually segmented thoracic soft tissue (heart + tumor) volume on a log scale show good corellation.

and 3 were significant for the soft tissue and lung. Notably, there were no statistically significant changes in the results between the 0 and 1 categories. Overall, the MLAST outputs had a Spearman's rank correlation when compared with the manual scores of -0.69, 0.51, and 0.32 for lung, soft tissue, and intermediate tissue, respectively.

## Comparison of MLAST to manual segmentation

We have shown that contrast-enhanced manual segmentation can produce quantitative tumor measurements [13] but is not a high-throughput method. We performed a side-by-side comparison of manual segmentation (Heart + Tumor volume) and corresponding MLAST-based segmentation (Soft + Intermediate tissue volume) from 10 non-contrast microCT scans in each qualitative manual scoring category, selected to ensure a wide distribution of tumor volumes. MLAST results were comparable to manual segmentation with a few notable distinctions (Fig 4B). Neither quantification approach indicated a statistically significant difference between the 0 and 1 manual scoring categories (which is not surprising with an ordinal scoring where 1 represents small, typically <5 mg, tumors and within MLAST signal to noise ratio (SNR)), though both demonstrated differences between higher categories. The overall relationship between the manually segmented volumes (slope = 0.7786, $R^2$ = 0.8939, p < 0.0001) demonstrated a good correlation between the two methods, but with a considerable y-offset and a slope of <1. Analysis of the Sorenson-Dice similarity coefficient demonstrated a mean overlap of 85.43 +/- 6.0% (n = 29) between MLAST and manual segmentations of the lung field.

## Comparison of MLAST to contrast-enhanced manual segmentation

Images from non-contrast microCT scans do not differentiate tissues with subtly different densities or vascularization. Vascular contrast agents such as Exitron nano-12000 can provide additional tissue contrast to differentiate between tumor and soft tissue [13]. To further demonstrate MLAST's quantitation capability we compared MLAST-based segmentation (Soft

+ Intermediate tissue volume) and manual segmentation (Heart + Tumor volume) values from non-contrast microCT scans with manual segmentation of contrast-enhanced microCT images from the KL-GEM model of lung cancer (Fig 5A). MLAST showed a strong regression trend (R = 0.9252, p < 0.0001) compared with contrast-enhanced segmentation, with a similar slope (<1) and offset (>0) when compared to manual segmentation of non-contrast scans (Fig 5A). Interestingly, manual segmentation of non-contrast scans showed an even stronger relationship to contrast-enhanced segmentation (R = 0.9874, p < 0.0001) (Fig 5A). MLAST segmented volume at low tumor burden was higher than the manual segmentation. This is mainly due to the fact that MLAST segmentation includes the normal anatomy (heart, esophagus, and

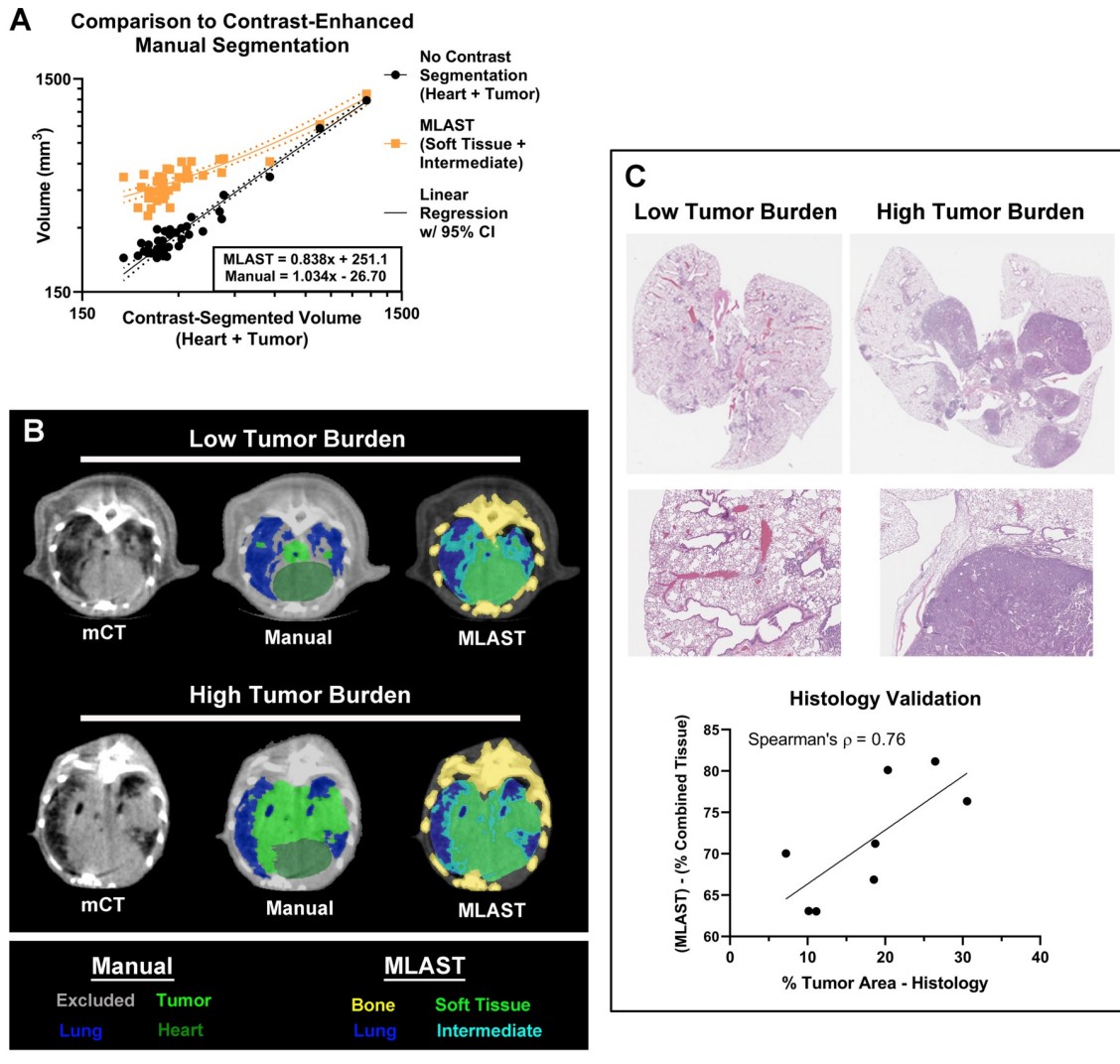

**Fig 5. Comparison of MLAST using contrast-enhanced microCT imaging and histology. (A)** Validation results showing linear regression of MLAST and manually segmented volumes vs contrast-enhanced manual segmentation on a log scale. The volumes of MLAST-segmented non-contrast scans, which are made up of soft tissue and intermediate, are shown in orange. The volumes of manually segmented non-contrast scans, which are made up of heart and tumor, are shown in black. The x-axis represents the volumes of manually segmented contrast-enhanced scans, which are also made up of heart and tumor. Regression lines (solid) are shown for both results, along with a 95% Confidence Interval (dotted). **(B)** Representative scans with low and high tumor burden by manual and MLAST segmentation. **(C)** MLAST validation with histology: Representative H&E stained images with low and high tumor burden at low and high magnification. The graph shows the correlation of tumor burden evaluation by MLAST to H&E methods.

vasculature), which was excluded in the manual segmentation. Fig 5B shows the segmentation of lung field with low- and high-tumor burden from non-contrast microCT scans. Analysis of the non-contrast scans using a Sorenson-Dice similarity coefficient demonstrated a mean overlap of 85.40 +/- 3.0% (n = 8) between MLAST and manual segmentations of the lung field.

As a final validation step, MLAST measurements were compared with whole slide analysis of H&E images of 2D lung sections (Fig 5C). Histology is a validated alternative method to evaluate lung tumor burden either as correlative endpoint or if microCT imaging is not available in research facilities [13, 22]. We observed robust correlation between the % of tumor area measured from H&E images and MLAST measurement of the % of combined tissue, which includes the soft tissue and intermediate (Spearman's correlation coefficient = 0.76).

## Evaluating therapeutic efficacy in KL-GEM model

To showcase the utility of the MLAST tool in pre-clinical studies, we evaluated the efficacy of a known therapeutic drug in KL-GEM model of lung cancer. microCT imaging was performed before initiation of treatment (baseline) and after 3 weeks of treatment with the drug. The microCT scans were evaluated by manual scoring and MLAST analysis (Fig 6). Although manual scoring showed an improvement in the treatment group over the control group, this difference was not statistically significant (Fig 6A). In comparison to manual scoring, MLAST lung measurements demonstrated a larger difference between baseline and post-treatment scans (Fig 6B). The lung percentage in the vehicle-treated control group decreased by ~30%, whereas in the drug-treated group it increased by ~15%. Non-contrast microCT images showed an increased tumor burden in the vehicle-treated control compared to the drug-treated group (Fig 6C).

## Discussion

Our analyses show that the MLAST algorithm is capable of accurately detecting major changes in the tumor burden of lung microCT images from GEMM mice. The comparison of MLAST segmentations to manual scoring demonstrated that a decrease in segmented lung percentage correlates with an increase in tumor burden and that major differences in tumor burden produce statistically significant differences in segmented lung percentage. Although MLAST may not be ideally suited for detecting small, individual tumor nodules in the early stages of tumor development due to the lack of significant difference in lower scoring categories, it offers significant advantages over manual scoring when evaluating lungs with a heavy tumor burden and significant therapeutic effects. Readouts from MLAST are quantitative and objective, while manual scoring is time-consuming and is susceptible to both intra- and inter-analyst variability. In addition, MLAST offers improved quantification with higher precision to scan evaluation than captured by manual scoring (for example, between 2 and 3), allowing for improved sensitivity of tumor burden quantification. This effect is potentially useful at lesion margins where the analyst is unable to distinguish between proliferating tumor edges and other lesions such as tumor-compression induced atelectasis [23], peritumor edema associated with pseudoprogression [24], or at tumor-organ interfaces (as shown in Fig 5B).

MLAST is not the first automation tool for lung tissue segmentation. There have been other efforts made over the last few years to automate image analysis for oncology models, including non-small cell lung cancer. Much of the work has been done in clinical areas, including both semi-automated [25] and fully-automated [26, 27] approaches. More recently, researchers have worked to bring deep learning and neural networks to the world of CT lung analysis, some focusing on classification [28] and others on segmentation [29, 30]. While these developments are important for the field of clinical imaging, segmentation in pre-clinical imaging comes with its own unique set of challenges. Small-animal lung CT has considerable inherent motion and

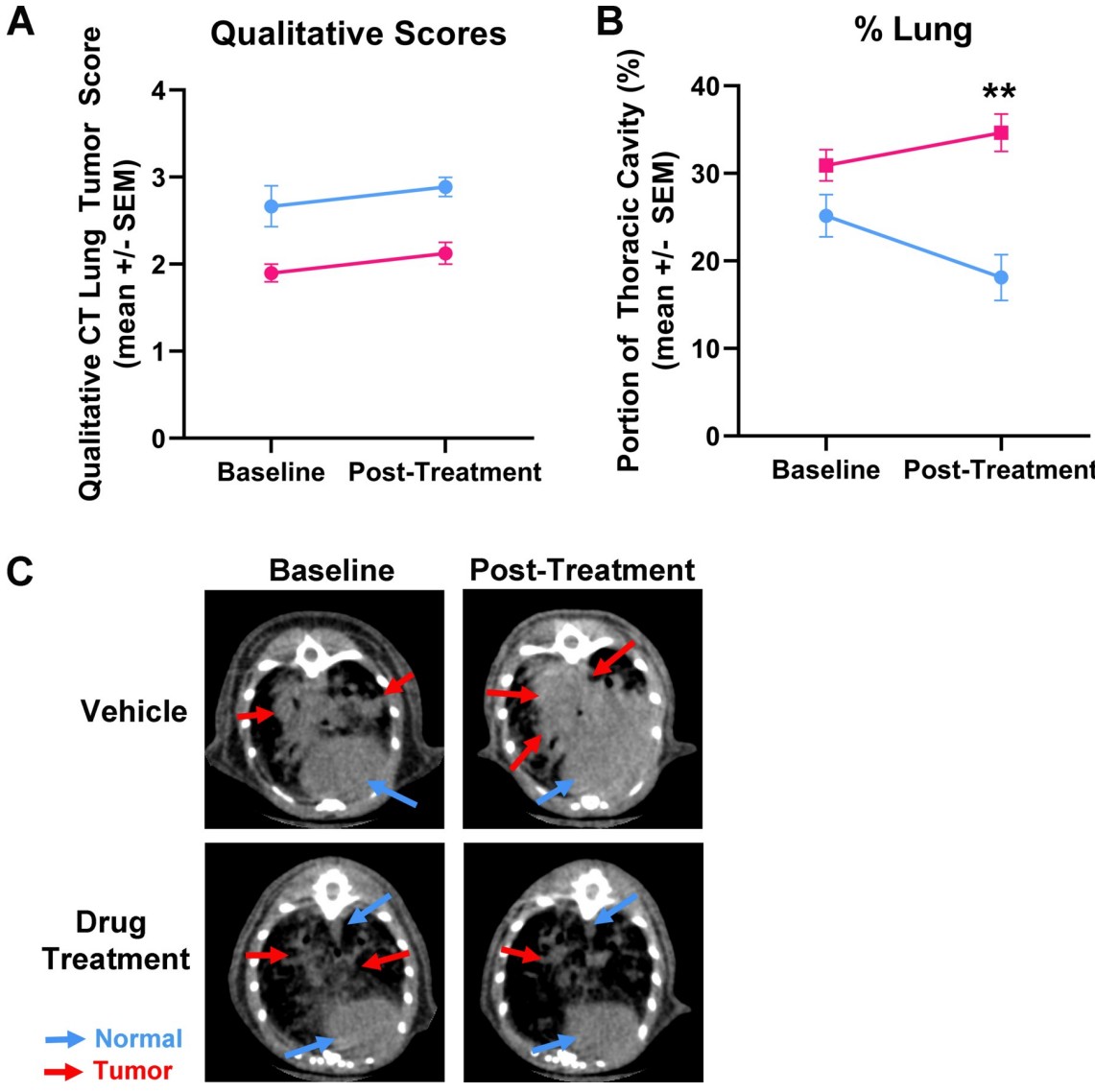

**Fig 6. Utility of MLAST in an efficacy trial using Kras/Lkb1 GEM model. (A)** Manual scores assigned to scans from vehicle control (blue) and drug treatment (pink) groups before (baseline) and 3 weeks post-treatment regimen. **(B)** MLAST scores from vehicle (blue) and drug treatment (pink) groups before and post-treatment regimen. **(C)** Representative images from vehicle control and drug treated groups before and post-treatment. Images show profound progression of tumor in vehicle control group, but not in drug treated group. (n = 10/group; +/- SEM; ** = p-value < 0.0001).

breathing artifact. Either prospective or retrospective respiratory gating can mitigate this problem, but both are time-consuming and often limit study throughput. Despite these challenges, several groups have developed semi-automated or even rules-based fully-automated algorithms for segmenting small animal lung microCT scans [25, 28, 29, 31, 32]. Haines et al. used a semi-automated threshold and region-grow algorithm to segment the combined tumor and vasculature (T&V) from the lung space with a correlation of $R^2 = 0.79$ to histology results [15]. More recently Barck *et al.* developed an algorithm to perform fully-automated T&V segmentation in GEMM mice with a correlation to manual metrics in the range of r = 0.83–0.93 [14].

MLAST is an adaptation of the Barck approach with several improvements: it uses automated thresholding by k-means rather than pre-defined values and it employs an intra-scan

normalization, both of which are designed to allow more flexibility. This design can accommodate variations in reconstruction parameters and sensitivity related to position in the gantry, as well as drifts in signal calibration. MLAST also requires minimal preprocessing and can be used across different scanners. Its flexibility allows the tool to be used on a diverse range of GEM models, orthotopic tumors, and non-oncology applications with gradations of density such as fibrosis [33] and pneumonia [34]. We have also refined the Barck mask function by stacking multiple slices before applying the spline function, which smooths the transition between slices and eliminates the inclusion of intercostal muscles. Finally, the diaphragm detection algorithm used by MLAST allows for inclusion of tumors at the lower lung region near the diaphragm that may otherwise be difficult to segment.

Comparison with manual segmentation volumes demonstrated the precision of tissue segmentation by MLAST. Tissue volumes calculated by automated segmentation compared well with those derived from manual segmentation, underscoring reliability of the algorithm for computation of tissue volumes in the thoracic cavity. Interestingly, neither MLAST nor manual segmentation was able to differentiate between scans scored 0–1, likely due to the relatively low impact of small tumors on the overall tissue volume in the thoracic cavity. The non-zero y-offset in the MLAST-segmented volumes is largely due to the volume of normal anatomy (heart, esophagus, and vasculature) which was included in the MLAST segmentation but not in the manual segmentation. Additionally, the vasculature is more visible in low-burden than high-burden scans where it merges with tumors, contributing to a regression slope of less than one, as the difference in volume between the MLAST segmentations and manual segmentations lessened in the higher-burden scans where the vasculature was merged with the tumor lesions (which is only distinguishable with contrast enhancement [35]).

Contrast-enhanced manual segmentation is considered a "gold standard" for quantitative CT imaging [13]. Thus, we compared MLAST measurements with manual segmentation of contrast-enhanced microCT images from the KL-GEM model. The high correlation between the manually-segmented tissue volumes with and without contrast indicates that our manual segmentations in both experiments were consistent and accurate relative to ground truth. The experiment confirmed that MLAST tissue segmentations were well correlated with manual segmentations and demonstrated its precision against the gold standard of contrast-enhanced manual lung segmentation.

The implementation of MLAST on a study to evaluate the anti-tumor effects of a therapeutic on KL-GEM tumors demonstrated the power of MLAST to detect treatment response. The quantitative measurements determined with MLAST allowed for detection of subtle differences within scoring categories, as the ordinal label of 3 does not seem to have fully captured the extent to which the tumors grew in the control group. Thus, MLAST was able to detect volumetric changes in tumors in the treatment group that could not be found by the manual scoring system. There is no significant change in either lung volume or overall thoracic volume with age in non-tumor bearing mice within our typical experimental time window [36]. Therefore we assume no age-related effects. This study provides an example of how MLAST has much potential value to the process of drug development as a tool to enhance sensitivity and increase quantitative power over ordinal binning in studies with moderate to heavy tumor burden.

Apart from quantitation and sensitivity, MLAST also offers efficiency. It can analyze a scan in approximately 1/5th of the time required for manual scoring (data not shown), and this speed can be further improved by running the algorithm on multiple CPUs at once. Entire studies could potentially be analyzed in only a few minutes without being monitored by a scientist, allowing for greater efficiency, faster data turnaround time, and more frequent intermediate readouts in longitudinal studies. MLAST could therefore enable drug development scientists to make decisions in near-real time and collect more frequent datapoints.

These benefits of a fully-automated, quantitative approach could be expanded even further in the future with the adaptation of a deep learning model for segmentation of pre-clinical lung CTs. If a training library of enough pre-segmented images could be generated, a U-Net model could be trained to segment tumor region specifically, rather than relying on quantification of the loss of lung tissue. It is reasonable to assume that a direct segmentation of the tumor itself would have an improved accuracy in tracking disease progression and would allow for further possibilities of analysis of those tumor nodules. Such an approach would be largely beneficial and should be pursued in the future.

The flexibility, ease-of-use, and efficiency of MLAST make it a valuable analytical asset for pre-clinical efficacy studies involving lung cancer as well as potentially for other models of pulmonary pathology. Automated analysis can potentially be applied to many other imaging paradigms to perform similarly quantitative, efficient analyses in a variety of bio-pharmaceutical therapeutic areas.

## Supporting information

**S1 Video. Full 3D flythrough of a representative lung from a tumor-bearing mouse show the segmentation from MLAST algorithm.**
(AVI)

**S1 Table. Raw minimal data for Fig 4A.** Manual scoring and MLAST results are shown for all subjects used in comparing manual scoring to MLAST segmentation. Manual scores are shown on a 0–3 scale and MLAST results are shown as % of total thoracic tissue.
(XLSX)

**S2 Table. Raw minimal data for Fig 4B.** Manual scores, manual segmentation volumes, and MLAST segmentation volumes are shown for scans used in comparing manual segmentation to MLAST segmentation. Volumes were calculated from pixel counts using an isometric pixel size of 100 μm, and are shown in $mm^3$.
(XLSX)

**S3 Table. Raw minimal data for Fig 5A.** Non-lung tissue volumes are shown as computed using manual segmentation in contrast-enhanced scans, manual segmentation in non-contrast scans, and MLAST segmentation in non-contrast scans for subjects at various time points post tumor induction. All volumes are in $mm^3$.
(XLSX)

**S4 Table. Raw minimal data for Fig 5C.** Tumor burdens as calculated by histology and MLAST are shown for the subjects used in comparing MLAST to contrast-enhanced manual segmentation.
(XLSX)

**S5 Table. Raw minimal data for Fig 6.** Manual scoring and MLAST results are shown for vehicle and treatment subjects at week 0 (pre-treatment) and week 3 (post-treatment). Manual scores are shown on a 0–3 scale and MLAST results are shown as % of total thoracic tissue.
(XLSX)

## Acknowledgments

Comparative Medicine, Pfizer Inc., for care of the animals, technical support, and funding. We thank Julita Ramirez (Comparative Medicine–Global Science and Technology, Pfizer Inc.) for editing the manuscript.

MLAST Tool can be accessed at: https://www.protocols.io/private/
12BD8BC1A93211EB938A0A58A9FEAC02

## Author Contributions

**Conceptualization:** John David, Anand Giddabasappa.

**Data curation:** Mary Katherine Montgomery, John David, Anand Giddabasappa.

**Formal analysis:** Mary Katherine Montgomery, John David, Sripad Ram, Shibing Deng, Vidya Premkumar, Lisa Manzuk, Anand Giddabasappa.

**Investigation:** Mary Katherine Montgomery, Haikuo Zhang, Shibing Deng, Vidya Premkumar, Lisa Manzuk, Ziyue Karen Jiang, Anand Giddabasappa.

**Methodology:** Mary Katherine Montgomery, John David, Haikuo Zhang, Sripad Ram, Shibing Deng, Vidya Premkumar, Lisa Manzuk, Ziyue Karen Jiang.

**Resources:** Haikuo Zhang.

**Software:** Mary Katherine Montgomery, John David, Vidya Premkumar.

**Supervision:** John David, Anand Giddabasappa.

**Validation:** Mary Katherine Montgomery, John David, Sripad Ram, Ziyue Karen Jiang.

**Visualization:** Sripad Ram.

**Writing – original draft:** Mary Katherine Montgomery, Anand Giddabasappa.

**Writing – review & editing:** Mary Katherine Montgomery, John David, Ziyue Karen Jiang, Anand Giddabasappa.

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
