## [Decision Letter · Decision Letter 0]

16 Feb 2021

Pécs, Hungary

February 15, 2021

PONE-D-20-38790

Mouse Lung Automated Segmentation Tool for Quantifying Lung Tumors after Micro-Computed Tomography

PLOS ONE

Dear Dr. Giddabasappa,

Thank you for submitting your manuscript to PLOS ONE. After careful consideration, we feel that it has merit but does not fully meet PLOS ONE’s publication criteria as it currently stands. Therefore, we invite you to submit a revised version of the manuscript that addresses the points raised by the Reviwers, listed below.

We look forward to receiving your revised manuscript.

Kind regards,

Joseph Najbauer, Ph.D.

Academic Editor

PLOS ONE

Journal Requirements:

2. We note in your Data Availability statement you have advised "No - some restrictions will apply". Could you please  clarify the nature of these restrictions, ie. if due to ethical or legal reasons.

PLOS defines a study's minimal data set as the underlying data used to reach the conclusions drawn in the manuscript and any additional data required to replicate the reported study findings in their entirety. All PLOS journals require that the minimal data set be made fully available. For more information about our data policy, please see http://journals.plos.org/plosone/s/data-availability.

3. Thank you for providing the following Funding Statement: 

"This project was funded by Pfizer, Inc."

We note that one or more of the authors is affiliated with the funding organization, indicating the funder may have had some role in the design, data collection, analysis or preparation of your manuscript for publication; in other words, the funder played an indirect role through the participation of the co-authors.

If the funding organization did not play a role in the study design, data collection and analysis, decision to publish, or preparation of the manuscript and only provided financial support in the form of authors' salaries and/or research materials, please review your statements relating to the author contributions, and ensure you have specifically and accurately indicated the role(s) that these authors had in your study in the Author Contributions section of the online submission form. Please make any necessary amendments directly within this section of the online submission form.  Please also update your Funding Statement to include the following statement: “The funder provided support in the form of salaries for authors [insert relevant initials], but did not have any additional role in the study design, data collection and analysis, decision to publish, or preparation of the manuscript. The specific roles of these authors are articulated in the ‘author contributions’ section.”

If the funding organization did have an additional role, please state and explain that role within your Funding Statement.

Please also provide an updated Competing Interests Statement declaring this commercial affiliation along with any other relevant declarations relating to employment, consultancy, patents, products in development, or marketed products, etc.  

Reviewers' comments:

Reviewer's Responses to Questions

**Comments to the Author**

1. Is the manuscript technically sound, and do the data support the conclusions?

Reviewer #1: Partly

Reviewer #2: Partly

2. Has the statistical analysis been performed appropriately and rigorously? 

Reviewer #1: N/A

Reviewer #2: Yes

3. Have the authors made all data underlying the findings in their manuscript fully available?

Reviewer #1: No

Reviewer #2: Yes

4. Is the manuscript presented in an intelligible fashion and written in standard English?

Reviewer #1: Yes

Reviewer #2: Yes

5. Review Comments to the Author

Reviewer #1: In this paper, the authors describe a new method that enables quick automatic segmentation of lung tumor in small animal models that might have use in drug studies. Experiments seem to have been conducted properly and results are interesting. However, the statements done regarding the results obtained are over rated and more self-critic is recommended.

My comments follow:

(1) Line 42: Please define what the intermediate region is. There is no definition of it in the text and it must be given in detail.

(2) Lines 107 and 109: Attention to the subscripts for carbon dioxide.

(3) Line 134: is it resolution or pixel size? The images suggest it is pixel size, not resolution.

(4) Line 105: it is Feldkamp, not Fledkamp.

(5) Lines 144-147: what is the size of a “small tumor nodule”? Precise values for “small”, “medium”, and “large” are needed here. An explanation is given in the legend of Fig 1 and more should be written in the main manuscript text.

(6) Line 128 and 279: KL-GEM or KL GEM?

(7) Line 148: write Amira 6.3 (ThermoFischer...).

(8) Line 160 - 162: please give a precise definition of a “significant decrease in the percentage of lung volume” must be given. As it is written, it is too imprecise.

(9) Line 205: Use “h”, not “hours”.

(10) Line 230: Title seems to be incomplete. It is a comparison of what to what?

(11) Figure 4 is hard to understand. Fig 4A: it is said it is a comparison of manual score with MLAST, but only the results of manual segmentation are shown (or only MLAST? The figure and the legend apparently do not match). Fig 4B: Why heart + tumor volume manually estimated should have a correlation with MLAST-based estimation of soft tissue + intermediate? Maybe this is obvious for the authors, but is not written in the text. Again, the same for Fig. 5. A and B. A comparison of manual segmentation to MLAST segmentation is the core of the paper, tough no explanation is given for why this comparison (heart + tumor volume vs. soft tissue + intermediate) makes sense scientifically and is the best that can be achieved in the context of this work. Then in Fig 5 C, the comparison of MLAST and histology only considers the tumor area for histology and soft tissue + intermediate for MLAST. Why is there a different comparison for histology?

(12) Line 250: I believe there’s something wrong with this sentence.

(13) Line 261: correct “side-by-side”.

(14) Line 279: a definition of KL-GEM model is needed in the methods part.

(15) Line 287: “Figure 5B shows the segmentation of lung. The results of manual and MLAST segmentation are obviously different and this different has to be addressed in the text. Of course, manual and MLAST segmentation are going to be different, but are the differences relevant? If not, why? The authors are interested in the statistical differences between the groups studied, but these differences in segmentation cannot be overlooked and the minimum number of scanned animals to give relevant results should be given.

(16) Line 296 and 303 (and other parts in the text): micro-CT or mCT? Choose one or other, because using both terms shows lack of consistency.

(17) Line 302: what is vehicle-treated group? Is it the control group? Clear information in the methods part is needed and the use of the term “control group” is recommended.

(18) Line 318: What is “finer granularity to scan evaluation”? A clear definition must be given.

(19) Line 320: a comparison of the images in Fig 3 (segmented-top and original-bottom) shows that MLAST overestimates the boarder of the bones. From the images, I cannot say for sure if the same happens with boarders within soft-tissue, but a rough visual analysis suggests unclear boarders between tissue types. Thus, I definitely do not agree with “This effect is potentially useful at lesion margins where the analyst is unable to distinguish between proliferating tumor edges and other lesions such as tumor compression induced atelectasis [22], peritumor edema associated with pseudoprogression [23], or at tumor-organ interfaces (as shown in Figure 5B).” More explanation and less speculation are needed here. Also, maybe the authors should consider presenting Fig 3 as Fig 5 B (superposition of the segmented areas in transparent colors on the micro-CT slice).

(20) Line 327: I recommend to add more references to the “semi-automated and fully-automated approaches”.

(21) Lines 357 and 370: the same statement is given twice, at the beginning and at the end of the paragraph: “Comparison with manual segmentation volumes demonstrated the accuracy of tissue segmentation by MLAST”. Once should be enough. However, as commented before, I see a problem in this comparison and, thus, this statement is questionable. A proper explanation and scientific basis for this comparison must be given.

(22) Line 373: “compared”, not “correlated”.

(23) Line 357-358: I do not agree with “Comparison with manual segmentation volumes demonstrated the accuracy of tissue segmentation by MLAST.” Fig 5 C shows some statistical correlation in the measurements, but does not really show accuracy. Is it possible to show figures such as Fig. 4 in https://doi.org/10.1038/s41598-018-37394-w? This would show accuracy. More self-critic is recommended.

(24) Line 382 - 384: Why is MLAST a “dynamic measurement”? Are the “subtle differences within scoring categories” relevant or they are an error of the method? How to tell the difference?

(25) Line 389-392: I agree that MLAST “has much potential value to the process of drug development”. It for sure can increase provide quick results that, within different populations, has a statistical relevance. However, I do not agree that the results indicate that it is more sensitive than the other methods used.

(26) Legends of Fig. 4 and 5: The use of the word “validation” is questionable. The figures show a comparison of the results, but do not show a proof, or a confirmation, as the word “validation” suggests. Use of “comparison” is more appropriate.

Reviewer #2: The authors describe a workflow to automatically analyze the lungs of mice that were imaged using µCT. This work describes a practical solution to a task in pharmaceutical research that should inspire the community to use processes that accelerate the search of new treatments against cancer.

Tumor growth will change the lung tissue composition and will become visible in the x-ray images. Based on the setup described here, tumor growth should give a strong signal and this would allow different ways for an analysis such as visual classification, or a densitometric analysis (i.e. shifts in voxel intensity histograms), or a size measurement after lung segmentation.

The authors use a size measurement after lung segmentation and describe the steps needed for a robust and fully automized procedure.

The results are compared to an analysis based on ground truth segmentation generated by human experts, a tumor burden score also performed by a human expert, and readouts generated from histology. The correlation proves the usability of the authors approach.

In Fig 2A a histogram of density values is shown. Histogram analysis seems to be a common tool in x-ray image analysis and would probably give information about the health state of lungs. The authors should discuss why they did not use the densitometry data to also analyze tumor presence or have they tried this with unsatisfactory results.

Fig 5A shows the weak part of the MLAST approach. Small tumor nodules in lung lobes that can easily be seen and used in a scoring scheme, can also be marked in a manual segmentation, but will have little effect on volumetric measurements. The smoothing effect of the clustering in MLAST leads to a significant offset. In the discussion the authors explain in detail all the effects caused by the different analysis methods used.

However neuronal networks are not used for this work. The authors argue that a deep learning approach to classify or segment the lung was not feasible because of missing ground truth in the public domain and therefore not considered. For the evaluation of their MLAST procedure they generated over 3500 ground truth images for segmentation and used 446 lungs for scoring, a source for ten thousands of images for a classification training. In the case of tumor burden in lungs this is sufficient material for an application using deep learning. The authors should rethink their arguments.

This manuscript describes in great detail a practical solution to a segmentation task. Alternative solutions that might even be easier and faster, are not discussed, but should be worth a few sentences.

6. PLOS authors have the option to publish the peer review history of their article (what does this mean?). If published, this will include your full peer review and any attached files.

Reviewer #1: No

Reviewer #2: No

---

## [Author Response · Author response to Decision Letter 0]

3 May 2021

Responses to Reviewers in in the Cover Letter/Response to Reviewers document.

---

## [Decision Letter · Decision Letter 1]

26 May 2021

Pécs, Hungary

May 25, 2021

Mouse Lung Automated Segmentation Tool for Quantifying Lung Tumors after Micro-Computed Tomography

PONE-D-20-38790R1

Dear Dr. Giddabasappa,

We’re pleased to inform you that your manuscript (R1 version) has been judged scientifically suitable for publication and will be formally accepted for publication once it meets all outstanding technical requirements.

PLEASE SEE THE COMMENT BY REVIEWER #2 BELOW (point 6).

Kind regards,

Joseph Najbauer, Ph.D.

Academic Editor

PLOS ONE

Reviewers' comments:

Reviewer's Responses to Questions

**Comments to the Author**

1. If the authors have adequately addressed your comments raised in a previous round of review and you feel that this manuscript is now acceptable for publication, you may indicate that here to bypass the “Comments to the Author” section, enter your conflict of interest statement in the “Confidential to Editor” section, and submit your "Accept" recommendation.

Reviewer #2: All comments have been addressed

2. Is the manuscript technically sound, and do the data support the conclusions?

Reviewer #2: Yes

3. Has the statistical analysis been performed appropriately and rigorously? 

Reviewer #2: N/A

4. Have the authors made all data underlying the findings in their manuscript fully available?

Reviewer #2: Yes

5. Is the manuscript presented in an intelligible fashion and written in standard English?

Reviewer #2: Yes

6. Review Comments to the Author

Reviewer #2: I recommend using the term microCT or µCT, but not mCT. m is usually considered milli and so the short form would be milliCT.

7. PLOS authors have the option to publish the peer review history of their article (what does this mean?). If published, this will include your full peer review and any attached files.

Reviewer #2: No

---

## [Editor Report · Acceptance letter]

8 Jun 2021

PONE-D-20-38790R1 

Mouse Lung Automated Segmentation Tool for Quantifying Lung Tumors after Micro-Computed Tomography 

Dear Dr. Giddabasappa:

I'm pleased to inform you that your manuscript has been deemed suitable for publication in PLOS ONE. Congratulations! Your manuscript is now with our production department. 

Kind regards, 

on behalf of

Dr. Joseph Najbauer 

Academic Editor

PLOS ONE